# Alpha-1 Antitrypsin Reduces Disease Progression in a Mouse Model of Charcot-Marie-Tooth Type 1A: A Role for Decreased Inflammation and ADAM-17 Inhibition

**DOI:** 10.3390/ijms23137405

**Published:** 2022-07-03

**Authors:** Nikolay Zhukovsky, Marianna Silvano, Thierry Filloux, Sergio Gonzalez, Karl-Heinz Krause

**Affiliations:** 1Neurix SA, Av. de la Roseraie 64, 1205 Geneva, Switzerland; 2Triskel Integrated Services SA, Ch. du Pavillon 5, 1218 Geneva, Switzerland; tfilloux@triskel.com; 3In Vivex SAS, Av. Louis Lumière 177b, 34400 Lunel, France; info@invivex.com; 4Faculty of Medicine, Department of Pathology and Immunology, University of Geneva, Ru. Michel-Servet 1, 1211 Geneva, Switzerland; karl-heinz.krause@unige.ch

**Keywords:** CMT1A, hereditary motor and sensory neuropathy (HMSN), recombinant AAT, interleukin-6 (IL-6), MHC class II (MHCII), peripheral myelin protein (PMP22)

## Abstract

Charcot-Marie-Tooth disease type 1 (CMT1A) is a hereditary peripheral neuropathy for which there is no available therapy. Alpha-1 antitrypsin (AAT) is an abundant serine protease inhibitor with anti-inflammatory and immunomodulating properties. Here, we tested whether treatment with human AAT (hAAT) would have a therapeutic effect on CMT1A in a *PMP22* transgenic mouse model. Our results show that hAAT significantly improved compound muscle action potential and histopathological features and decreased circulating IL-6 in CMT1A mice. We also investigated some of the possible underlying mechanisms in vitro. We confirmed that hAAT inhibits ADAM-17, a protease that has been implicated in blocking myelination. Furthermore, both hAAT and recombinant human AAT (rhAAT) were able to attenuate the activation of a macrophage/microglia cell line, markedly decreasing the activation of the MHC class II promoter and the expression of pro-inflammatory genes such as *IL-1β* and the endoplasmic reticulum (ER) stress marker *ATF3*. Taken together, our results demonstrate for the first time that hAAT is able to reduce the progression of CMT1A, possibly by dampening inflammation and by regulating ADAM-17. Given the already well-established safety profile of hAAT, specifically in AAT deficiency disease (AATD), we suggest that the findings of our study should be promptly investigated in CMT1A patients.

## 1. Introduction

Alpha-1 antitrypsin (AAT) is an acute phase protein that increases in concentration in response to inflammation. Produced mainly in the liver by hepatocytes, but also in monocytes, neutrophils, macrophages, microglia, epithelial cells, and pulmonary alveolar cells, it is the most abundant serine protease inhibitor in human plasma. In AAT deficiency disease (AATD), the elastin and collagen of the lung alveoli are not sufficiently protected from neutrophil elastase, a serine protease, which results in progressive destruction of lung parenchyma, emphysema, and chronic obstructive pulmonary disease (COPD). Besides liver cirrhosis due to the accumulation of mutated hAAT in the hepatocytes, few other morbidities are seen in AATD. Intravenous infusion of plasma derived human alpha-1 antitrypsin (hAAT) was approved by the United States Food and Drug Administration (FDA) to treat emphysema associated with AATD, and there are now several FDA-approved therapeutic products on the market, all of which are derived from pooled human blood plasma and have similar efficacy and safety profiles [1,2].

In experimental disease models, it has been recognized that hAAT has a much larger function as an anti-inflammatory and immunomodulating compound that cannot be explained by its serine protease inhibitor activity alone. These properties have been shown in animal models of human disease, for instance graft-versus-host disease, inflammatory bowel disease, rheumatoid arthritis, acute liver failure, and type 1 diabetes. In peripheral neuropathies, an important hallmark is the defective production of myelin sheaths surrounding the nerves, which results in reduced transmission of action potentials. In the peripheral nervous system (PNS), the myelin is produced by Schwann cells (SCs) by a complex synthesis that is partly dependent on a disintegrin and metalloprotease 17 (ADAM-17), which is able to block the myelination of axons through ligand-independent cleavage of neuregulin-1 subtype III (Nrg1-III). This results in peripheral nerves that are insufficiently protected by myelin and lack a protective sheath, a defect present in many peripheral neuropathies and in Charcot-Marie-Tooth disease type 1A (CMT1A). CMT1A is a hereditary neuropathy that arises due to duplication of the *PMP22* gene on chromosome 17p11.2 [3], which in turn leads to overexpression of a 22 kDa protein called peripheral myelin protein, also known as PMP22, a key factor for maintaining the cohesiveness and elasticity of the myelin sheath [4]. The inability of SCs to form a myelin sheath (dysmyelination) in CMT1A is believed to be associated with the accumulation of PMP22 protein in the endoplasmic reticulum (ER), which subsequently overwhelms the cells and leads to apoptosis. In addition, the lack of myelination may have its origins in a macrophage-related inflammation process [5]. CMT1A patients have also been shown to have increased numbers of peripheral lymphocytes, indicating that activated lymphocytes might participate in the loss of myelin [6].

Unfortunately, at present there are no therapies available for any type of CMT [4]. Herein, we provide the first evidence on the potential therapeutic role of hAAT in CMT1A.

## 2. Results

### 2.1. Effects of Human Plasma Derived AAT (hAAT) in PMP22 Mice

This study evaluated the neuromuscular performance of *PMP22* mice treated subcutaneously with hAAT at 50 mg/kg twice daily for 14 days. The effects of treatment were also measured using sciatic nerve conductivity and histology. hAAT plasma levels, interleukin-6 (IL-6), and tumor necrosis factor alpha (TNFα) were measured during treatment to ensure systemic exposure to hAAT and to measure the changes in these cytokines during the study.

#### 2.1.1. Plasma hAAT Levels

Treatment with hAAT commenced on day 14. hAAT was detected in the plasma of the CMT1A + hAAT group at mean values of 6.07, 6.99, 8.15, and 5.22 µg/mL on days 14, 19, 24, and 29, respectively, and consistent systemic exposure to hAAT during treatment was shown (hAAT values (mean ± SD) were 6.07 ± 1.05, 6.99 ± 0.56, 8.15 ± 0.91, and 5.22 ± 0.68, respectively).

#### 2.1.2. Neuromuscular Tests

At 6 weeks of age (day 14), the baseline values of rotarod latency to fall (time the mouse stays on the rotarod) and the grip strength test measured before the initiation of treatment were not significantly different across groups. At 8 weeks of age, vehicle-treated mice exhibited impaired neuromuscular performance, as shown by significantly decreased rotarod latency and mean grip strength relative to the wild-type animals. While the hAAT-treated mice presented slightly increased rotarod latency and mean grip strength compared to the vehicle-treated group, this increase was not statistically significant due to the small number of animals (Appendix A).

#### 2.1.3. Sciatic Nerve Electrophysiology

Baseline values at 6 weeks were not significantly different across groups for compound muscle action potential (CMAP) amplitude and nerve conduction velocity (NCV). As expected, strong and significant decreases in CMAP amplitude and NCV were observed in the CMT1A + vehicle group compared to the wild-type control group at 8 weeks old. Importantly, mice receiving the hAAT treatment presented a significant increase in CMAP amplitude compared to the CMT1A + vehicle-treated group (Table 1 and Figure 1), suggesting a protective effect of hAAT against axonal degeneration. hAAT-treated mice showed a slight although non-statistically significant increase in NCV compared to CMT1A + vehicle (Appendix A).

#### 2.1.4. Sciatic Nerve Histology

As expected, a significant decrease in the total number of axons per surface and the axonal diameter and a significant increase in the g-ratio were observed in the CMT1A + vehicle group compared to the wild-type control group at 8 weeks old (Table 2). A slight but nonsignificant increase in the total number of axons per surface was observed in the CMT1A + hAAT-treated group compared to the vehicle group (Appendix A). Importantly, a significant increase in the axonal diameter and decrease in the g-ratio (ratio of inner to outer diameter of a myelinated axon) were observed in the CMT1A animals treated with hAAT compared to the vehicle-treated group. The CMT1A + hAAT mice also presented a statistically different axonal diameter and g-ratio from the wild-type control group (Table 2 and Figure 2, Figure 3 and Figure 4). Taken together, the data suggest a positive but partial effect of hAAT on the histopathology induced by CMT1A when administered subcutaneously at 50 mg/kg twice daily.

#### 2.1.5. Plasma Cytokine Levels

Similar plasma IL-6 concentrations were observed between groups at baseline (day 1, 4 weeks old) and on day 8. As the CMT1A phenotype progressed, a significant increase in plasma IL-6 concentration was observed in the CMT + vehicle group on day 14 (first day of treatment) and day 29. Animals treated with hAAT also presented significantly increased plasma IL-6 concentration compared to baseline on day 14. On day 29, vehicle-treated animals had increased plasma IL-6 levels, while animals treated with hAAT had only slightly increased levels relative to day 14, which was significantly lower compared to vehicle-treated animals at this time (Table 3 and Figure 5), thus suggesting a direct or indirect effect of hAAT on this inflammatory cytokine.

A significant increase in plasma TNFα concentration was observed in the CMT1A + vehicle group on days 14 and 29. Treatment with hAAT had no effect on TNFα levels in this model of CMT1A, as evidenced by a significant increase in plasma TNFα concentration observed on days 14 and 29 compared to the baseline concentration, which was not significantly different from that observed in the CMT1A + vehicle group (Appendix A).

### 2.2. AAT Inhibits Drivers of Dysmyelination and Inflammation In Vitro

#### 2.2.1. AAT Inhibition of ADAM-17 Activity

In order to clarify the nature of hAAT action on myelin restoration, we decided to first evaluate the effect of hAAT on ADAM-17, since it has been shown to regulate myelin production in Schwann cells [4] and there are controversial results on hAAT and ADAM-17 interaction [7,8].

Here we show that hAAT can significantly inhibit ADAM-17 activity in a dose-dependent manner in cell-free conditions (Figure 6A,B), suggesting that hAAT might act through the inhibition of ADAM-17.

#### 2.2.2. AAT Reduces MHCII Activation in Inflammatory Cells

Since inflammatory immune cells can contribute to demyelinating neuropathy [9], we decided to investigate mechanisms that may be related to AAT activity in an in vitro inflammatory model.

Macrophage/microglia cells (HMC3) stably transduced with MHCII luciferase promoter and inflamed with interferon gamma (IFNγ) were chosen to produce an inflammatory model. First, we induced inflammation with IFNγ at different doses for 24 h (inflamed cells), and we checked the level of MHCII activation through luciferase activity. At 10 ng/mL, IFNγ was already able to increase MHCII luciferase activity five-fold, therefore we chose this concentration for the next set of experiments (Figure 7).

Next, we evaluated the effect of AAT on this model. Cells were treated with either human plasma derived AAT (hAAT) or recombinant human AAT (rhAAT) or transduced with *UbiAAT:GFP* lentivector, and the latter was sorted by GFP to select cells with 50% or 100% AAT expression. MHCII luciferase activity increased in cells treated with IFNγ only. It decreased in a dose-dependent manner in cells treated with AAT (plasma derived and recombinant) and in cells expressing *UbiAAT:GFP* (Figure 8A,B). Our results show that AAT can strongly reduce MHCII activation in inflamed cells, and this is in line with its anti-inflammatory activity.

#### 2.2.3. Transcriptomic Characterization of Inflammatory Cells Treated with AAT

Inflamed and non-inflamed cells were treated with plasma-derived and recombinant AAT and then challenged for transcriptomic analysis, after being tested for their MHCII luciferase activity (Appendix A).

Gene set enrichment analysis (GSEA) confirmed that the inflammatory profile induced by IFNγ was induced by upregulation of several processes related to inflammation (Appendix A).

AAT treatment in the absence or presence of IFNγ was able to significantly modulate some of these pathways (Figure 9, Appendix A). Importantly, AAT significantly impaired the IFNγ-induced inflammatory response related to the pathology and progression of neurodegenerative/demyelinating diseases [10]. 

AAT treatment counterbalances the expression of inflammatory genes as shown in tables of genes related to antigen presentation (Table 4 and Appendix A), cytokine signaling (Table 5 and Appendix A). A substantial number of inflammatory genes (~35%; top of Table 4) were found to be affected by AAT treatment (6 of 23 genes related to antigen presentation, 14 of 33 genes related to cytokine signaling), showing the anti-inflammatory potential of AAT. *HLA-DOB* and *HLA-DRA* (Table 4) genes were induced in inflammation and downregulated by AAT, which is consistent with the reduction in MHCII activity after AAT treatment (Figure 8). AAT decreases the expression of important inflammatory cytokines such as *IL-1β* (Table 4 and Table 5), as already reported in another experimental setting [11].

Taken together, the data suggest that in an inflammatory context, AAT can attenuate MHCII activity, thus affecting different biological responses, such as IFNγ related responses and inflammation.

## 3. Discussion

Our study shows that: (i) hAAT has a positive effect on compound muscle potential in CMT1A mice, indicating partial recovery of axonal degeneration; (ii) hAAT can ameliorate histopathological signs of CMT1A; (iii) hAAT can reduce long-term inflammatory response in vivo (plasma IL-6); and (iv) hAAT mode action can be driven by different molecular mechanisms related to ADAM-17 inhibition, MHCII repression, and inflammatory response modulation.

hAAT has anti-inflammatory properties and the ability to protect cells from apoptosis, as reported in several studies [11,12,13]. Our RNAseq data indicate that *IL-1β* expression increased in inflamed cells and subsequently decreased in inflamed cells treated with AAT, suggesting that the neuroinflammatory process may involve the *NLRP3* inflammasome [14]. Interestingly, *ATF3*, which is one of the most important players in the ER stress response and is activated in the CMT1A mouse model [15,16], was upregulated in inflammation and downregulated after AAT treatment.

The curative and beta-cell regenerative effects of hAAT were demonstrated in an autoimmune mouse model of type 1 diabetes [12] as well as inhibition of apoptosis in pancreatic beta-cells [17]. The unfolded protein response (UPR) plays an important role in CMT1A, and PMP22 aggregates have been detected in a CMT1A mouse model and in patients [18,19]. AAT treatment of inflamed cells upregulates the UPR, which may attenuate pro-apoptotic signaling. Interestingly, UPR attenuation has recently been tested in a pre-clinical study of CMT1A mice, highlighting the importance of the UPR in the progression of the disease [20]. Moreover, in our in vitro experimental setting, the *p53* pathway and genes related to ER stress (e.g., *ATF3*) were downregulated by AAT, suggesting that AAT may be involved in modulating these biological responses in CMT1A as well.

hAAT was previously shown to reduce axon demyelination and secreted IL-6 levels in a mouse model of EAE-induced multiple sclerosis [13], which is consistent with the effects observed in our CMT1A mouse model.

The absence of a myelin sheath (dysmyelination) is a major hallmark of CMT1A. Myelin production is regulated by several pathways. ADAM-17 blocks the myelination of axons by SCs by cleaving the epidermal growth factor domain of Nrg1-III [4]. Contrasting evidence has been reported on the ability of hAAT to inhibit ADAM-17 [7,8]. Our results show that hAAT is indeed able to inhibit ADAM-17, confirming the original report by Bergin et al. (2010), elucidating one possible mechanism of action for hAAT in the context of CMT1A.

It has been shown that macrophages of CMT1A patients had increased expression of MHCII [21]. Macrophages have also been implicated in myelin disruption, and accordingly, macrophage-associated demyelination was observed in peripheral nerve biopsies of CMT1A patients [5,9]. MHCII expression is typically induced by IFNγ released from lymphocytes [22]. Thus, the upregulation of MHCII in CMT1A patients argues in favor of an immune-driven inflammatory response [6]. hAAT was able to restore the compound muscle action potential (and hence axonal function) in our CMT1A model in vivo, and to repress MHCII-induced activation in inflammatory cells in vitro. This suggests that in peripheral neuropathies such as CMT1A, hAAT has the potential to protect myelin integrity by inhibiting MHCII expression in macrophages.

Future in vitro and in vivo studies are needed to better elucidate the multifaceted mode of action of hAAT in CMT1A for a more complete understanding. Importantly, hAAT may at last offer CMT1A patients a therapeutic avenue targeting the underlying etiology of the disease rather than simply addressing symptoms that are often too divergent or broad in nature. The relevance of the CMT neuropathy score (CMTNS) can be further improved by objective measurements in humans in addition to the electrophysiological recording of the compound muscle action potential (CMAP) to also include plasma levels of IL-6 and MRI [23,24]. 

Strikingly, the genetic prevalence of AATD is on par with that of CMT1A, which is estimated to affect 30 in 100,000 individuals or around 2.4 million people worldwide. At the same time, hAAT has the potential to become an important remedy for CMT1A and other neuropathies with similar pathological modalities. 

As such, recombinant human AAT (rhAAT) could be important in meeting the increased demand for hAAT replacement and replenishment therapies. To that end, we show that rhAAT produced in a CHO cell line is as potent as that derived from plasma (hAAT) in ameliorating IFNγ-induced inflammation in inflammatory cells (Figure 8B). Even though CMT1A is not a life-threatening condition, the onset of the disease is already present in childhood, and it progresses in severity with age. CMT1A is a disability that affects millions of people worldwide, throughout the duration of their lives, and there are still no effective therapeutic options. hAAT offers a promising therapeutic avenue for progressing from these pre-clinical findings to clinical trials without delay. 

Our study provides strong evidence, both in vitro and vivo, for a protective role of hAAT in CMT1A pathology. It highlights the strength of the CMT1A animal model in quantitatively assessing the pro-myelinating properties of hAAT. Employing RNAseq to study macrophage/microglia cells has proven to be a powerful method for identifying some of the fundamental drivers of the disease state at the gene expression level.

Yet, this study has weaknesses and limitations:Even if the CMT1A mouse model is a well-designed model that mimics many aspects of human disease, additional work with human material will be important. In particular, in vitro disease modeling using patient-derived stem cells could be an excellent approach for furthering our understanding of the mode of action of AAT in CMT1A [25]. As inflammation is difficult to model in vitro, mice with a humanized immune system might be an interesting alternative. As such, the CMT1A mutation would need to be introduced into NOD/SCID mice.The sample size of our animal studies was relatively small (*n =* 3). Nevertheless, hAAT significantly improved muscle action potential and histopathological features, and decreased circulating IL-6 in CMT1A mice; virtually all other parameters of disease severity showed a trend toward improvement. Thus, the effect of hAAT is strong and can be recognized even with a small sample size. However, additional in vivo studies are needed. Given the high potential for rapid clinical translation of hAAT therapy, such studies should be designed as part of a preclinical package.Our in vitro experiments provide initial clues to the molecular mechanism behind the protective effect of AAT. However, further studies will be needed for a more complete picture of the cascade of interactions involving AAT in CMT1A. We only investigated changes on the mRNA level. The importance of the proteasome and protein misfolding by impaired Schwann cells in demyelinating disease [26] suggests that additional proteomic research will be necessary to elucidate post-transcriptional mechanisms in CMT1A pathophysiology.The fine balance between the onset of inflammation and subsequent damage to the myelin sheath remains poorly understood and clearly needs further study. This is a promising direction for the discovery of novel therapeutic targets, in our opinion.Importantly, the therapeutic effect of AAT might not be limited to CMT1A alone, and could probably also be extended to other genetic, inflammatory, metabolic, or toxic neuropathies. For example, chronic inflammatory demyelinating polyneuropathy (CIDP) could also possibly be treated with hAAT, alone or in combination with immunoglobulin (IgG antibody) therapy.

Further studies will be needed to investigate these points.

In summary, our study provides initial convincing evidence for a possible role of hAAT in the management of CMT1A and possibly other peripheral neuropathies. We also provide initial insights into the potential underlying mechanism of action of AAT in this context. Given the multiple morbidities associated with CMT1A and the proven excellent tolerability of hAAT in AATD, we believe that clinical studies should be rapidly initiated.

## 4. Materials and Methods

### 4.1. Animals and Treatment

*PMP22* transgenic mice (*B6.Cg-Tg(PMP22)C3Fbas/J*, In vivex housing service (Montpellier, France)) were received at 3 weeks of age and housed in pairs in disposable Makrolon cages (M-BTM, Innovive (San Diego, CA, USA)) with filter hoods, at a constant temperature with a day/night cycle of 12/12 h. Animals received water (M-WT-300, Innovive (San Diego, CA, USA)) and nutrition (A03, SAFE (Rosenberg, Germany)) ad libitum. The ethics approval number for this study is D3417223, APAFIS#23920-2020020320279696.

For the experiment, 3 groups of 3 animals each were assigned to the following subcutaneous treatment twice daily at 7:00 A.M. and 7:00 P.M. for 2 weeks, starting from 6 weeks and up to 8 weeks of age (Figure 10):Wild-type (WT) mice receiving vehicle, 0.9% NaCl (positive control)*PMP22* mice receiving vehicle, 0.9% NaCl (negative control)*PMP22* mice receiving hAAT (A9024 and A6510, Sigma-Aldrich) at 50 mg/kg per injection

To avoid any interference from the mouse anti-human antibody response elicited by hAAT treatment, the treatment duration was set to 2 weeks, and it was started at 6 weeks of age, when the clinical signs become measurable. The subcutaneous route was selected to avoid the potentially fatal anaphylaxis related to sharp increases in plasma levels of hAAT, which can be observed using the intraperitoneal route [27]. Intradermal administration also leads to delayed peak plasma hAAT levels; however, the small injection volume allowed by this route precluded its use.

The dose was selected based on previous publications [27,28]. For this preliminary study, the daily dose was set to 100 mg/kg, which was split into 2 doses of 50 mg/kg administered at 7:00 A.M. and 7:00 P.M.

Plasma sampling and ELISA measurements: Starting from 4 weeks of age (study day 1), animals were subjected to blood sampling to determine the plasma levels of cytokines and hAAT. The tail vein was punctured with a 20 G lancet and 80 μL of blood was collected in a heparinized microtube. Samples were centrifuged for 15 min at 1000× *g* (or 3000 rpm) at 2–8 °C within 30 min of collection, and 45 μL of supernatant (plasma) was stored at −20 °C before ELISA analysis. IL-6 and TNF-α levels were measured on days 1, 8, 14, and 29 using commercial ELISA kits (Ref. RAB0308 and Ref. RAB0477, respectively; Sigma-Aldrich). Plasma levels of hAAT were evaluated every 5 days from the first to the last days of treatment (days 14 and 29) using a commercial ELISA kit (Abcam Ref. ab189579), according to the manufacturer; species cross-reactivity was less than 3% with mouse AAT.

Rotarod test: On study days 14 and 29, mice were given a 1-day pre-training trial to familiarize them with the rotating rod. On days 15 and 30, latency to fall was measured at successively increasing speed from 4 to 40 rpm over a 300 s maximum time period. Each animal underwent 3 trials a day, and the values were averaged for each animal, and then for each group.

Grip strength test: Grip strength of all limbs was measured on study days 15 and 30 by supporting the limbs on a horizontal grid connected to a gauge and pulling the animal’s tail. The maximum force (newtons) exerted on the grid before the animal lost its grip was recorded, and the mean of 3 repeated measurements was calculated. Finally, the data were averaged for each treated group.

Sciatic nerve electrophysiology: Standard electromyography was performed on mice under ketamine/xylazine anesthesia on study days 15 and 30. A pair of steel needle electrodes (MLA1302; AD Instruments (Paris, France)) was placed subcutaneously along the nerve at the sciatic notch (proximal stimulation). A second pair of electrodes was placed along the tibial nerve above the ankle (distal stimulation). Supramaximal square-wave pulses lasting 10 ms at 1 mA were delivered using a PowerLab 26T (AD Instruments). Compound muscle action potential (CMAP) was recorded from the intrinsic foot muscles using steel electrodes. Both amplitude and latency of CMAP were determined. The distance between the 2 sites of stimulation was measured alongside the skin surface with fully extended legs, and nerve conduction velocity (NCV) was calculated from sciatic nerve latency measurements.

Sciatic nerve sampling and histology: On day 30, animals were sacrificed. The left leg skin was opened, and the full sciatic nerve was sampled. The left sciatic nerve of all animals was fixed for 20 min in situ with 4% PFA and 2.5% glutaraldehyde in 0.1 M phosphate buffer (pH 7.3). Then, nerves were removed and post-fixed overnight in the same buffer. After washing for 30 min in 0.2 M PBS buffer, the samples were dehydrated using ethanol gradient solutions and embedded in epoxy resin. Semi-thin cross-sections were cut and stained with 0.5% toluidine blue + 1% borax + 100 mL MilliQ water. The axonal diameter, number of myelinated motor axons, and myelin g-ratio were quantified using the ImageJ g-ratio plug-in.

### 4.2. Generation of Transgenic HMC3 Cells

HMC3 cells have been described as a valuable tool to study human microglial activation [29]. More recently, the HMC3:*MHCIILuc* cell line coding for Renilla luciferase under major histocompatibility complex II promoter (*HLA-DRA*) was generated through transduction with a lentiviral vector. This cell line was obtained through a tech transfer from the University of Geneva. 

HMC3-*MHCIILuc* cells expressing hAAT under the control of ubiquitin promoter (HMC3-*MHCIILuc:UbiAAT GFP* cells) were generated by Neurix. The lentivector coding for human AAT under ubiquitin promoter and green fluorescent protein (GFP) under human PGK promoter was obtained according to the protocol described in [30]. In brief, 4.5 × 106 HEK cells were plated in a Ø100 mm dish and transfected 16 h later with 15 μg of *pCWXPG-UBI-SP:AAT*, 10 μg of packaging plasmid (psPAX2, gift from Didier Trono; Addgene plasmid 12260), and 5 μg of envelope (pMD2G, gift from Didier Trono; Addgene plasmid 12259). The medium was changed 8 h post-transfection. After 48 h, the viral supernatant was collected and filtered using 45 μm PVDF filters and stored at −80 °C. HMC3-*MHCIILuc:UbiAAT* cell lines were sorted for GFP expression and 100% and 50% of positive cells were selected for experimental condition. 

### 4.3. Human Microglial Cell Line Culture 

These cell lines were cultured on TC-treated cell culture dishes (CELLSTAR^®^, Greiner, 7.664160) in high-glucose DMEM + glutamine (Gibco, 41965039) supplemented with 10% (*v*/*v*) fetal bovine serum (FBS, Gibco, 10270106) and 100 μg/mL penicillin/streptomycin (Pen/Strep, 15070063; Thermo Fisher Scientific, Waltham, MA, USA). Cultures were maintained at 37 °C in a 5% CO_2_ atmosphere. Passages were performed by quickly rinsing the cells in 1× PBS, with 3 min trypsinization at RT (Tryple Express, 12604021; Thermo Fisher Scientific, Waltham, MA, USA), followed by centrifugation (5 min, 1000 RPM) and resuspension in the supplemented DMEM. Cells were counted and plated at the desired concentration.

### 4.4. IFNγ-Mediated Human Microglia Activation

The HMC3-*MHCIILuc* cell line was seeded into 96-well plates at a density of approximately 2500 cells/well. After 24 h, their activation was induced with 24 h IFNγ (Sigma, St. Louis, MO, USA, SRP3058) presentation at a range of concentrations (0.1, 1, 10, or 100 ng/mL). IFNγ was then removed, and cells were cultured for 48 h before being assessed for cell viability and activation.

### 4.5. Exogenous/Endogenous AAT Treatment of IFNγ-Activated Human Microglia

HMC3*-MHCIILuc* and HMC3*-MHCIILuc:UbiAAT* (endogenous AAT) cell lines were seeded into 96-well plates at a density of approximately 2500 cells/well. The HMC3- *MHCIILuc* cell line was plated. Plasma-derived and recombinant AAT (AAT 1 and 2, produced in CHO cells) were added 3 h later at a range of concentrations (1, 10, or 25 μM). After a further 24 h, still in the presence of exogenous or endogenous AAT, microglial activation was induced with 24 h IFNγ presentation (10 ng/mL). IFNγ was then removed and both the HMC3-*MHCIILuc* and HMC3-*MHCIILuc:UbiAAT* cells were cultured for 48 h in the presence of exogenous or endogenous AAT before cell cultures were assessed for cell viability and activation.

### 4.6. Measurement of Human Microglia Cell Viability and Activation

The viability (Cell Counting Kit-8, 96992; Sigma) and activation (Renilla-Glo^®^ Luciferase Assay System, E2710; Promega, Madison, Wisconsin, USA) of HMC3-*MHCIILuc* and HMC3*-MHCIILuc:UbiAAT* cell cultures were assessed according to the manufacturers’ protocols.

### 4.7. RNA Collection and Sequencing, and Analysis of Differential Expression

RNA extraction was achieved with RNeasy Mini kit (Qiagen, Hilden, Germany) according to the manufacturer’s protocol. RNA samples from plasma-derived AAT and recombinant AAT 2 were checked for quality (2100 Bioanalyzer, Agilent, Santa Clara, CA, USA) and libraries prepared with Truseq RNA Library Kit (Illumina, San Diego, CA, USA, RS-122-2001). Libraries were sequenced (HiSeq 4000, Illumina), controlled for the quality of sequencing (FastQC http://www.bioinformatics.babraham.ac.uk/projects/fastqc/ accessed on 16 May 2022), and mapped on the human genome (STAR v.2.7.0f; UCSC hg38); then, the reads were counted (HTSeq v0.9.1) and a differential expression analysis was performed with the R bioconductor package (edgeR 1.30.1. Robinson MD, McCarthy DJ and Smyth GK (2010). edgeR: a Bioconductor package for differential expression analysis of digital gene expression data. Bioinformatics 26, 139–140).

### 4.8. Cell-Free ADAM-17 Activity

ADAM-17 activity and its inhibition by human AAT (AAT) were measured with Recombinant Human ADAM-17 kit (930-ADB and ES003, R&D Systems, Minneapolis, MN, USA) in black 96-well immunoplates (437111; Thermo Fisher Scientific). The enzymatic activity of ADAM-17 was measured by mixing 0.005 μg of rhADAM-17 with 10 μM of Mca-PLAQAV-Dpa-RSSSR-NH2 fluorogenic peptide substrate III in assay buffer (25 mM Tris, 2.5 μM ZnCl_2_, 0.005% Brij-35 (*w*/*v*), pH 9.0) to a final volume of 100 μL. AAT (batch A6150; Sigma Aldrich) was resuspended in water (vehicle), and control ADAM-17 activity was assessed in the presence of the vehicle (amount used for AAT 100 μM). AAT was added at different concentrations (0, 6.25, 12.5, 25, 50, and 100 μM) to assess its dose-dependent inhibition of ADAM-17. All conditions were performed in triplicate. Activity was measured as relative fluorescent unit (RFU) in kinetic mode (9 time points over 5 min) with a SpectraMax iD3 Microplate Reader (low PMT gain, 1 s exposure, top read at 1 mm, wavelengths: excitation 320 nm, emission 405 nm).

## 5. Patents

A patent application resulting from the work reported in this manuscript has been filed with the European Patent Office.

## Figures and Tables

**Figure 1 ijms-23-07405-f001:**
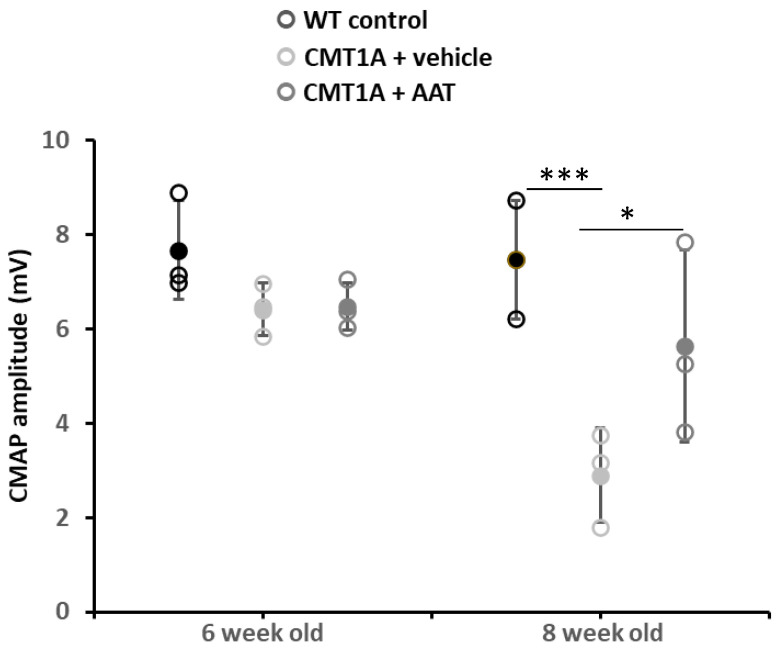
Compound muscle action potential. Graph represents data shown in Table 1. One-way ANOVA and Tukey test: *** *p* < 0.001 vs. WT control; * *p* < 0.05 vs. CMT1A + vehicle. CMT1A + AAT vs. WT control is not significant. In this and all in vivo experiments, *n* = 3 animals per group.

**Figure 2 ijms-23-07405-f002:**
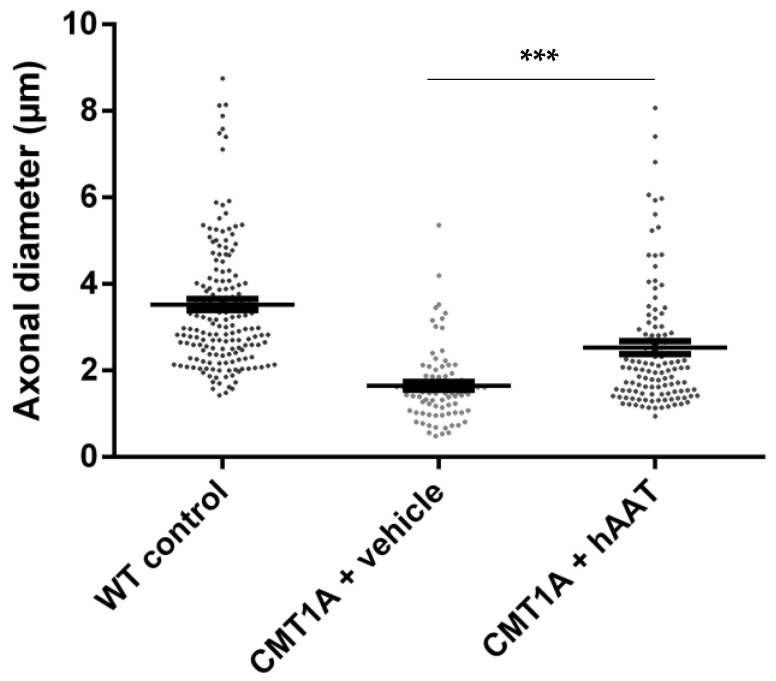
Axonal diameter. Graph represents data shown in Table 2. One-way ANOVA and Tukey test: *** *p* < 0.001 vs. WT control (not shown), *** *p* < 0.001 vs. CMT1A + vehicle.

**Figure 3 ijms-23-07405-f003:**
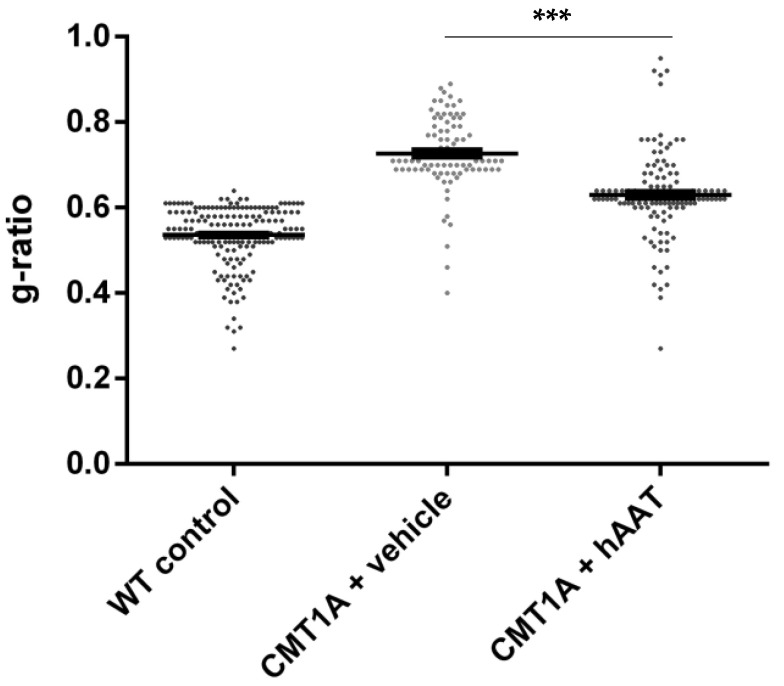
g-ratio. Graph represents data showed in Table 2. One-way ANOVA and Tukey test: *** *p* < 0.001 vs. WT control (not shown), *** *p* < 0.001 vs. CMT1A + vehicle.

**Figure 4 ijms-23-07405-f004:**
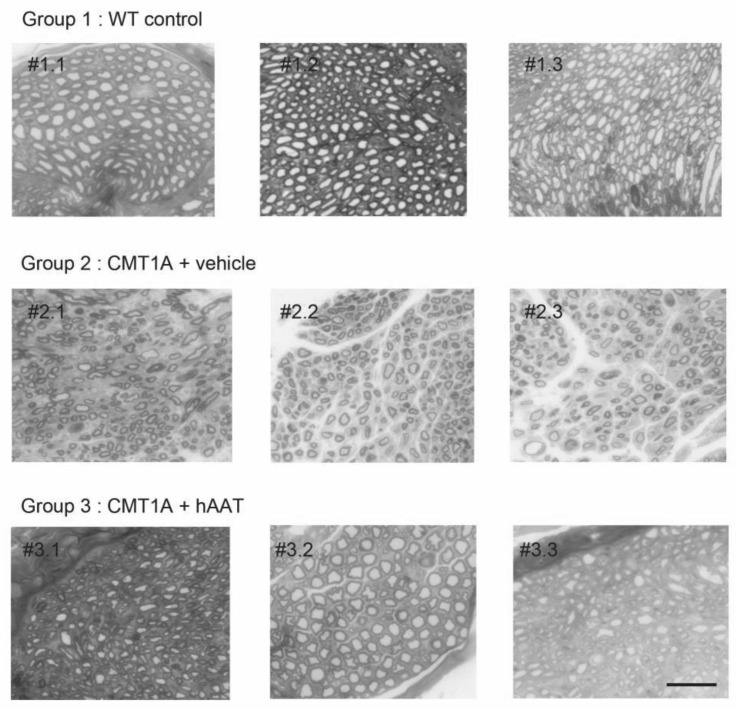
Individual histology images of semi-thin cross-sections of sciatic nerve. Each image represents one single animal. Toluidine blue staining was used to quantify number of axons and degree of myelination (see Figure 2 and Figure 3 for quantification). Degree of myelination was evaluated by axonal diameter and g-ratio. hAAT-treated group (group 3) showed improvement in these parameters compared to the vehicle group (Group 2). Scale bar = 10 µm. # = cross-section.

**Figure 5 ijms-23-07405-f005:**
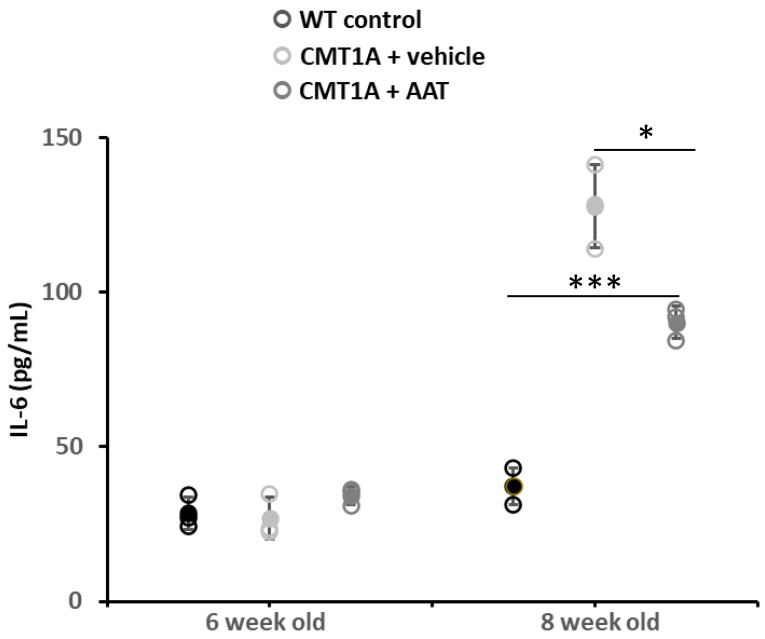
Plasma IL-6 concentration. At 29 days, CMT1A + vehicle mice showed statistically significantly increased IL-6 plasma concentration compared to control mice. On the contrary, plasma IL-6 was significantly downregulated in mice treated with hAAT compared to CMT1A, even if the level was still high compared to control mice. *t*-test: *** *p* < 0.001 compared to control, * *p* < 0.05 compared to CMT1A + vehicle. See also Appendix A.

**Figure 6 ijms-23-07405-f006:**
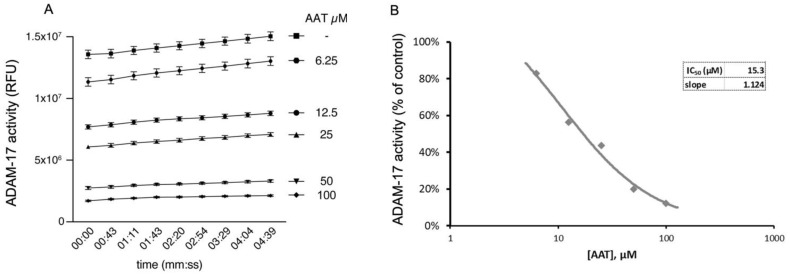
AAT inhibits ADAM-17 activity in cell-free assay, shown as (**A**) relative fluorescence unit over time (min) and (**B**) IC_50_, of 15.3 µM, representing concentration at which AAT exerts half its maximal inhibitory effect on ADAM-17 activity. All conditions were repeated in triplicate and are shown as mean ± SD. ANOVA: *p* < 0.01 for all conditions compared to control.

**Figure 7 ijms-23-07405-f007:**
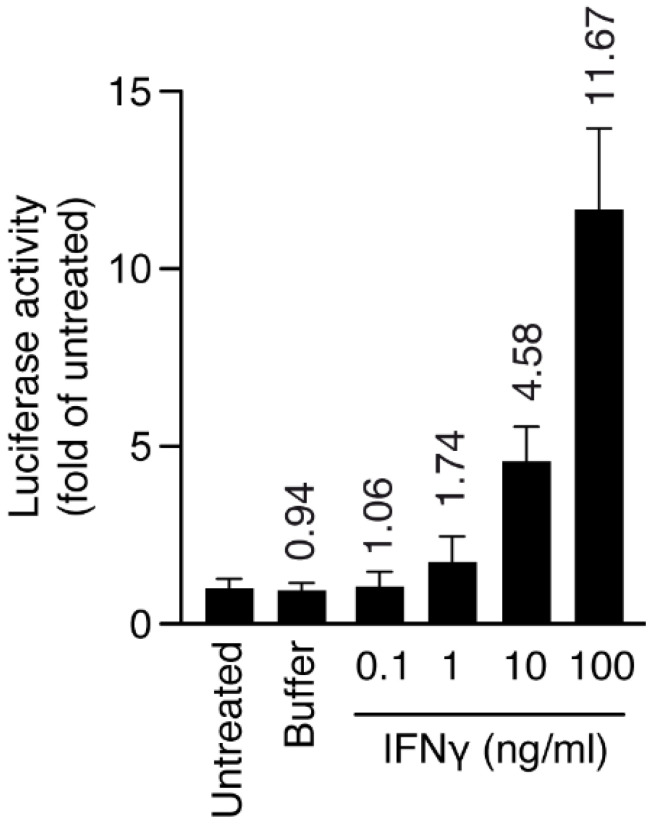
Human HMC3-*MHCIILuc* cells were plated at day 0 and activated with IFNγ for 24 h, and luciferase activity and cell viability were measured after 48 h. Numbers on top of bars represent fold increase over untreated cells. Luciferase activity was normalized over cell viability for all samples. Error bars are s.d. *p*-values by *t*-test: *p* < 0.05 for IFNγ (10 and 100 ng/mL).

**Figure 8 ijms-23-07405-f008:**
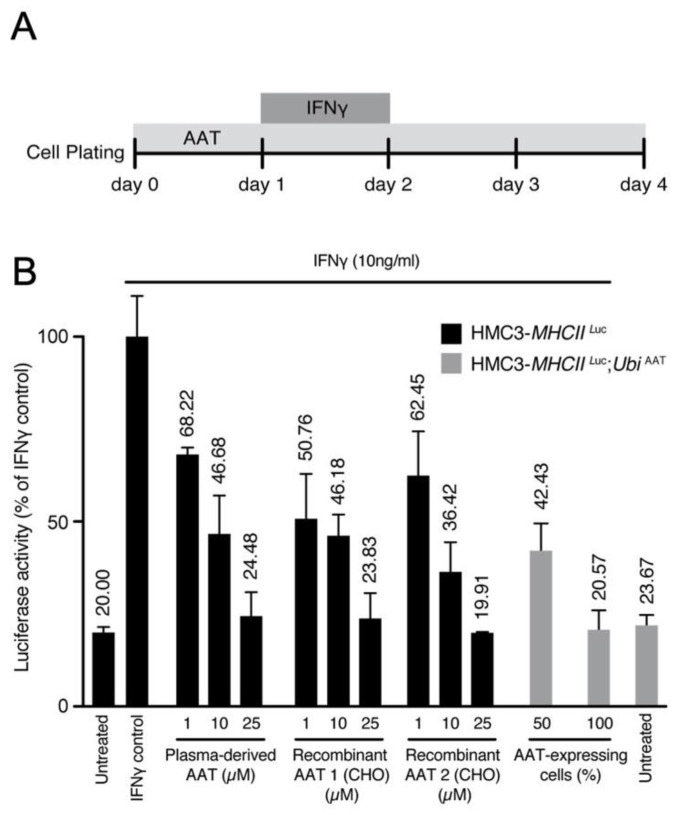
(**A**) Scheme for administration of IFNγ and AAT treatment. (**B**) MHCII activation was measured by luciferase and normalized to cell viability. Luciferase activity for all conditions is represented as fold change relative to control. A potential effect of the highest concentration of buffer used for drug presentation (IFNγ, AAT) was excluded. At 72 h after infection, human HMC3-*MHCIILuc* and HMC3-*MHCIILuc:UbiAAT* cells were sorted by GFP (50–100% expressing cells), and all treatments were performed as described in (**A**). All conditions were performed in triplicate; error bars represent standard deviation.

**Figure 9 ijms-23-07405-f009:**
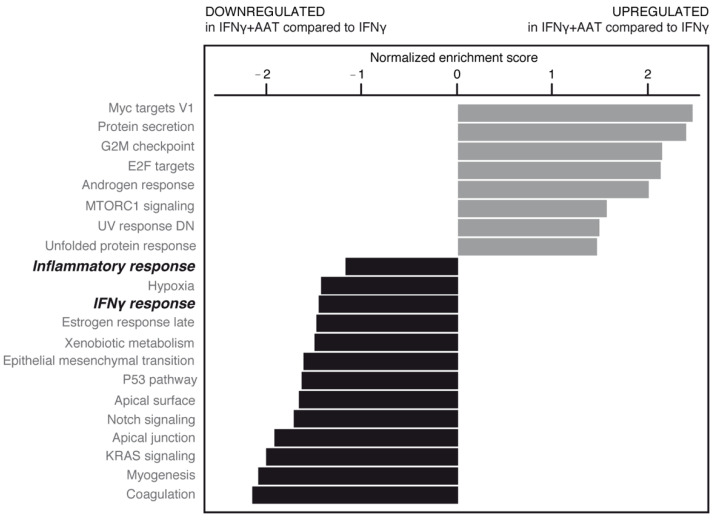
RNAseq data from plasma-derived and recombinant AATs were pooled, normalized, and analyzed by gene set enrichment analysis (GSEA) (shown here in inflamed cell condition). Upregulated (gray bars) and downregulated (black bars) gene families are shown according to normalized enrichment score. AAT was able to significantly modify some of these pathways (bold italic indicates pathways subjected to further analysis; see text).

**Figure 10 ijms-23-07405-f010:**
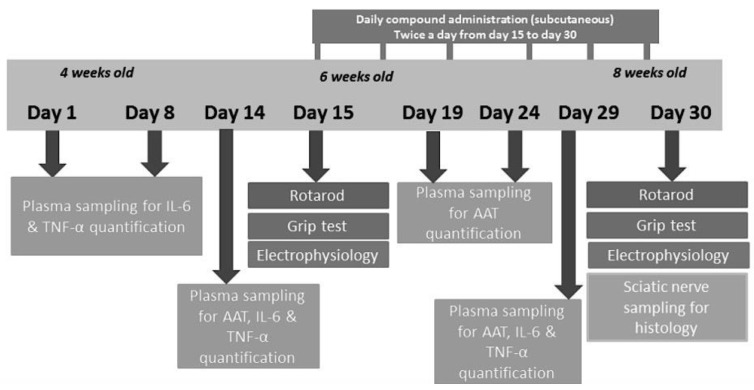
Study scheme.

**Table 1 ijms-23-07405-t001:** Effect of hAAT on compound muscle action potential and nerve conduction velocity. Mean ± SEM, *n* = 3 per group, 2-way ANOVA with repeated measures and Bonferroni *t*-test: ^†^ *p* < 0.05 vs. CMT1A + vehicle; *, *** *p* < 0.05, 0.001 vs. WT control.

	CMAP (mV)		NCV (m/s)	
	6 Weeks Old	8 Weeks Old	6 Weeks Old	8 Weeks Old
WT control	7.67 ± 0.61	7.46 ± 0.73	32.03 ± 0.76	30.59 ± 2.73
CMT1A + vehicle	6.41 ± 0.32	2.89 ± 0.58 ***	25.03 ± 1.42	12.68 ± 2.12 ***
CMT1A + AAT	6.47 ± 0.30	5.63 ± 1.18 ^†^	24.91 ± 4.13	20.89 ± 1.63 *

**Table 2 ijms-23-07405-t002:** Sciatic nerve histology. Mean ± SEM, *n* = 3 per group, 1-way ANOVA and Tukey test: ^†††^ *p* < 0.001 vs. CMT1A + vehicle, *** *p* < 0.001 vs. WT control. WT, wild type, * *p* < 0.05 CMT1A + vehicle vs. WT control.

	Axons/100 µm^2^	Axonal Diameter (µm)	G-Ratio
WT control	57.67 ± 0.88	3.52 ± 0.13	0.54 ± 0.005
CMT1A + vehicle	28.00 ± 7.02 *	1.65 ± 0.09 ***	0.73 ± 0.010 ***
CMT1A + AAT	39.33 ± 4.81	2.53 ± 0.15 ^†††^ ***	0.63 ± 0.009 ^†††^ ***

**Table 3 ijms-23-07405-t003:** Changes in plasma IL-6 levels with time following treatment with hAAT at 50 mg/kg b.i.d. or vehicle. Mean ± SEM, *n* = 3 per group; Student *t*-test: ^†^
*p* < 0.05 vs. CMT1A + vehicle at this time point; *, **, *** *p* < 0.05, 0.01, 0.001 vs. WT control at this time point. Day 14: first day of treatment. WT: wild type.

	Day 1	Day 8	Day 14	Day 29
WT control	28.50 ± 3.04	29.74 ± 2.67	35.90 ± 4.516	37.20 ± 3.391
CMT1A + vehicle	26.90 ± 3.86	38.32 ± 4.78	85.04 ± 2.296 ***	127.84 ± 7.810 ***
CMT1A + AAT	34.19 ± 1.67	39.10 ± 1.91 *	84.21 ± 4.727 **	90.20 ± 3.085 ^†^ ***

**Table 4 ijms-23-07405-t004:** Antigen presentation. Top inflammatory genes (FC > 2, *p*-value < 0.05; left column) were defined by differential expression between untreated and IFNγ-treated cells (inflammation; middle column). Top genes that were significantly and oppositely regulated by AAT treatment are shown in the right column.

Top Gene	*p*-Value	FC	*p*-Value	FC
	Up in inflammation		DOWN in inflammation + AAT	
*PSMB9*	5.73 × 10^−11^	5.96	3.11× 10^−2^	−1.23
*HLA-DOB*	5.11× 10^−4^	4.48	6.82× 10^−3^	−4.49
*HLA-DRA*	4.99× 10^−10^	4.76	4.68× 10^−2^	−2.22
*IL1B*	5.54× 10^−4^	2.08	1.11× 10^−3^	−2.07
	DOWN in inflammation		UP in inflammation + AAT	
*CALR*	6.15× 10^−11^	−3.53	4.74× 10^−4^	1.35
*PYCARD*	6.92× 10^−3^	−2.86	1.53× 10^−2^	2.20

**Table 5 ijms-23-07405-t005:** Cytokine signaling. Gene analysis was performed as described for Table 4.

Top Gene	*p*-Value	FC	*p*-Value	FC
	UP in inflammation		DOWN in inflammation + AAT	
*ATF3*	6.86× 10^−13^	7.218	1.85× 10^−3^	−1.26
*PSMB9*	6.09× 10^−10^	5.963	3.11× 10^−2^	−1.23
*FGF2*	1.66× 10^−6^	3.608	9.13× 10^−3^	−1.57
*CXCL1*	1.36× 10^−4^	2.82	7.22× 10^−3^	−2.09
*TNFRSF9*	2.39× 10^−4^	2.434	1.23× 10^−3^	−2.35
*CXCL5*	5.07× 10^−5^	2.36	5.98× 10^−3^	−2.15
*IL1B*	2.14× 10^−3^	2.08	1.11× 10^−3^	−2.07
	DOWN in inflammation		UP in inflammation + AAT	
*FSCN1*	9.42× 10^−8^	−6.312	8.09× 10^−4^	1.48
*IL4I1*	3.22× 10^−3^	−4.19	3.22× 10^−3^	1.95
*IL27RA*	2.22× 10^−7^	−3.848	2.67× 10^−2^	1.36
*STAT5A*	7.66× 10^−4^	−3.53	9.78× 10^−3^	1.45
*IL1R1*	1.27× 10^−7^	−2.566	3.94× 10^−2^	1.18
*JAK3*	3.03× 10^−8^	−2.43	3.20× 10^−3^	1.43
*RASAL1*	1.36× 10^−6^	−2.34	4.65× 10^−3^	1.52

## Data Availability

Transcriptomic data have been provided to the journal.

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
