# Peer review of "Alpha-1 Antitrypsin Reduces Disease Progression in a Mouse Model of Charcot-Marie-Tooth Type 1A: A Role for Decreased Inflammation and ADAM-17 Inhibition"

_ijms, 2022, doi:10.3390/ijms23137405_

Round 1

Reviewer 1 Report

The authors present results of effects of human alpha-1-antitrypsin in a specific mouse model of hereditary peripheral neuropathy. The results offer support for dose-finding studies in patients and a phase II randomized clinical trial.

Major comments:

1. AAT serum levels are reported in line 76. It is rather surprising that at day 29, the AAT level is 5.22 ug/ml. What is the SD belonging to all reported serum AAT levels? What is the explanation for such a long half-life in serum of AAT (which is very different from that in humans)?

2.In the legend to Table 1 (line 109) it is stated that p<0.001 vs WT control. Please add to which value(s) is belongs.

3. For the non-expert on sciatic nerve analysis, the legend of Figure 4 (line 141) is not explanatory. Each of the six images of mice in group 2 and 3 appear rather different. Please indicate in the legend the differences between vehicle and hAAT nerves.

Author Response

We would like to thank the editor and the reviewers for their positive feedback and helpful comments.

Please find below our point-by-point answer.

Reviewer 1

The authors present results of effects of human alpha-1-antitrypsin in a specific mouse model of hereditary peripheral neuropathy. The results offer support for dose-finding studies in patients and a phase II randomized clinical trial.

Major comments:

Q: AAT serum levels are reported in line 76. It is rather surprising that at day 29, the AAT level is 5.22 ug/ml. What is the SD belonging to all reported serum AAT levels? What is the explanation for such a long half-life in serum of AAT (which is very different from that in humans)?

A: The treatment has started at day 14, therefore, the level of hAAT at day 29 (after two weeks of treatment) is as expected. In fact, the low hAAT level at day 29 is due to the appearance of anti-drug antibodies (ADAs) at this time, as hAAT is an exogenous protein for mice. This has already been documented in the literature (Ma, 2010), where ADAs appear after 2 weeks of treatment. According to the GLASSIA U.S. Biologics License Application (BLA), the half life of hAAT in humans is about 110 hours. According to Baranovski et al. (2016), the half life of hAAT in C57Bl6 mice is about 9.5 hours, which is consistent with the human half life considering an allometric scaling factor of 12.1 for mice.

hAAT values (mean ± SD) were: 6.07 ± 1.05 at day 14; 6.99 ± 0.56 at day 19; 8.15 ± 0.91 at day 24 and 5.22 ± 0.68 at day 29 (see the text lines 86-87).

Q: In the legend to Table 1 (line 109) it is stated that p<0.001 vs WT control. Please add to which value(s) is belongs.

A: The legend of Table 1 was corrected as follows: *; ***: p<0.05; p<0.001 vs WT control (line 114).

Q: For the non-expert on sciatic nerve analysis, the legend of Figure 4 (line 141) is not explanatory. Each of the six images of mice in group 2 and 3 appear rather different. Please indicate in the legend the differences between vehicle and hAAT nerves.

A: The legend of Figure 4 was modified as follows:

Figure 4. Individual histology images of sciatic nerve semithin cross sections. Each image represents one single animal. Sciatic nerve toluidine blue staining was used to quantify the number of axons and the degree of myelination (see Figures 2 and 3 for the quantification). The degree of myelination was evaluated through axonal diameter and g-ratio calculation. hAAT treated group (Group 3) showed an increase of these parameters compared to the vehicle group (Group 2). Scale bar 10 µm (line 146).

The final manuscript will be sent to the Journal's English editing services before publication. 

Reviewer 2 Report

Dear Author,

This is a well-written scientific manuscript. This article addresses a highly important topic, especially in the context of Charcot-Marie-Tooth disease type 1 (CMTIA) peripheral neuropathy, which is without any available therapy. In the present work the authors demonstrated that AAT significantly reduced the progression of CMT1A. Authors also discuss additional ways ADAM-17 can support the fight against inflammation.

Below, please find minor suggestions:

KEYWORDS: The tittle words should not be repeated in Keywords

RESULTS

Line 87: At 6 weeks of age, the baseline values of the rotarod latency to fall??? and of the grip strength test were not significantly different across groups.

DISCUSSION

Line 308: Contrasting evidence  have has been reported in relation to the ability of hAAT to inhibit ADAM-17 [3, 24].

Please highlight the strengths and weaknesses (or limitations) of your methods and implications for researchers carrying out future studies in the same field.

Author Response

We would like to thank the editor and the reviewers for their positive feedback and helpful comments.

Please find below our point-by-point answer.

Reviewer 2

Dear Author,

This is a well-written scientific manuscript. This article addresses a highly important topic, especially in the context of Charcot-Marie-Tooth disease type 1 (CMTIA) peripheral neuropathy, which is without any available therapy. In the present work the authors demonstrated that AAT significantly reduced the progression of CMT1A. Authors also discuss additional ways ADAM-17 can support the fight against inflammation.

Below, please find minor suggestions:

Q: KEYWORDS: The tittle words should not be repeated in Keywords

A: We removed redundant words.

RESULTS

Q: Line 87: At 6 weeks of age, the baseline values of the rotarod latency to fall??? and of the grip strength test were not significantly different across groups.

A: The sentence was corrected to: “At 6 weeks of age (at day 14), the baseline values of the rotarod latency to fall (time the mouse stays on the rotarod) and of the grip strength test measured before the initiation of treatment, were not significantly different across groups.” (line 89).

DISCUSSION

Q: Line 308: Contrasting evidence  have has been reported in relation to the ability of hAAT to inhibit ADAM-17 [3, 24].

A: We corrected “have” with “has” (line 319).

Q: Please highlight the strengths and weaknesses (or limitations) of your methods and implications for researchers carrying out future studies in the same field.

A: We thank the reviewer for his/her comment, and we now discussed extensively strengths and limitations of our study in Discussion (line 363).

The final manuscript will be sent to the Journal's English editing services before publication.